# Timely initiation of antenatal care and associated factors among pregnant women attending antenatal care in Southwest Ethiopia

Toffik Redi[1], Oumer Seid[2], Getaw Walle Bazie[3]*, Erkihun Tadesse Amsalu[3], Niguss Cherie[4], Melaku Yalew[4]

1 Ministry of Health, Addis Ababa, Ethiopia, 2 Department of Nutrition and Dietetics, College of Medicine and Health Sciences, Bahir Dar University, Bahir Dar, Ethiopia, 3 Department of Epidemiology and Biostatistics, School of Public Health, College of Medicine and Health Sciences, Wollo University, Dessie, Ethiopia, 4 Department of Reproductive Health, School of Public Health, College of Medicine and Health Sciences, Wollo University, Dessie, Ethiopia

☯ These authors contributed equally to this work.
* getaw4jesus@gmail.com

## Abstract

### Background

The timing of initiation of first antenatal care visit is paramount for ensuring optimal care and health outcomes for women and children. However, the existing evidence from developing countries, including Ethiopia, indicates that most pregnant women are attending antenatal care in late pregnancy. Thus, this study was aimed to assess timely initiation of antenatal care and associated factors among pregnant women attending antenatal care services in Southwest Ethiopia.

### Methods

Institutional based cross-sectional study was conducted among 375 pregnant women from April 15 to June 15, 2019 in Southwest Ethiopia. A structured and pre-tested face-to-face interviewer-administered questionnaire technique was used to collect data. Systematic random sampling technique was employed to recruit pregnant women. The data were entered into Epi data version 4.4.2 and analyzed using SPSS version 25. Frequency tables, charts and measures of central tendency were used to describe the data. The effect of each variable on timely initiation of antenatal care was assessed using bi-variable logistic regression. A multivariable logistic regression model was used to identify factors associated with timely initiation of antenatal care. The adjusted odds ratio with 95% confidence interval and p<0.05 was used to identify factors associated with timely initiation of antenatal care.

### Results

The study revealed that 41.9% of pregnant women started antenatal care timely. Pregnant women who had good knowledge of timely initiation of antenatal care (AOR = 3.8, 95% CI:

**Funding:** The author(s) received no specific funding for this work.

**Competing interests:** The authors have declared that no competing interests exist.

**Abbreviations:** ANC, Antenatal Care; AHR, Adjusted Hazard Ratio; CHR, Crude Hazard Ratio; CI, Confidence Interval; EDHS, Ethiopian Demographic and Health Survey; HEW, Health Extension Worker; SPSS, Statistical Package for Social Sciences; WHO, World Health Organization.

2.2–6.5), planned to be pregnant (AOR = 5.1, 95% CI: 2.9–8.9), being primigravida (AOR = 2.6, 95% CI: 1.4–4.7) and confirmed their pregnancy by urine test (AOR = 4.1, 95% CI: 2.4–6.9) were found to be significant predictors for timely initiation of antenatal care.

## Conclusions

Despite the efforts made to make ANC visit services freely available, timely initiation of antenatal care among pregnant women in the study area was low. Pregnant women who had good knowledge of timely initiation of antenatal care, planned to be pregnant, being primigravida and confirmed pregnancy by urine test were found to be significant predictors for timely initiation of antenatal care. Therefore, efforts that strengthen awareness on antenatal care and its right time of commencement, increase pregnant women's knowledge of timing of antenatal care services and reducing unplanned pregnancies should be organized.

## Background

Woman's death during and following pregnancy and childbirth is a serious public health concern. About 295, 000 women died during and following pregnancy and childbirth in 2017 worldwide and 5.1 million babies are stillborn or die in first month of life [1] and 94% of all maternal deaths occur in low and lower middle-income countries [2]. Roughly two-thirds (196,000) of maternal deaths occurred in sub-Saharan Africa [3].

Women are screened for risk factors and receive appropriate advice, get tetanus toxoid vaccinations, health education and counseling on individual birth planning, intermittent presumptive treatment of malaria and iron supplementation during follow-up at the antenatal care (ANC) clinics. It is helpful to diagnose pre-existing health problems or to detect health complications. In turn, it leads to reduction in maternal mortality and morbidity [4].

Antenatal care from organized health care services has a paramount importance to ensure every pregnancy ends up with the delivery of a healthy child and keeping the health of the mother [5]. The World Health Organization (WHO) recommends that women and adolescent girls should have at least eight ANC contacts during pregnancy to improve perinatal outcomes and women's experience of care. The first visit is recommended to be during the first trimester [6].

Late initiation of antenatal care affects the health of both a child and the mother. Late initiation and inadequate utilization of ANC during pregnancy contributes to adverse maternal health outcomes such as maternal mortality [7]. Early initiation of ANC allows health professionals to treat and manage other treatable health conditions such as congenital anomalies, syphilis, control hypertension, anemia, control HIV/AIDS transmitted from mother to child and prevention of malaria complications that the woman may develop during pregnancy [8, 9].

Studies revealed that socio-demographic characteristics of pregnant women, past experience of ANC service utilization, parity, knowledge of the timing of ANC and pregnancy, and gender-biased cultures are the main factors that influence use of timely ANC in developing countries [10, 11]. In sub-Saharan Africa, low ANC coverage, low frequency of visits and late attendance at first antenatal visit were found to be common problems [12].

Ethiopia has been implementing maternal and newborn health interventions and has achieved significant improvements in the coverage. ANC services are being given free of charge in all governmental health institutions. However, only 20% of women had their first ANC during the first trimester [13]. Studies conducted in Ambo and Mekele, Ethiopia showed

that the timely initiation of ANC was 13.2% and 32.7%, respectively. Studies on timely initiation of ANC and its associated factors are conducted in different parts of Ethiopia [4, 7, 12, 14, 15]. However, there is scarcity of information on timely initiation and associated factors in the study area. In addition, specific empirical evidence is needed to tackle problems related to late initiation of antenatal care and improve the health of a child and the mother during and after pregnancy. Therefore, this study was aimed to determine timely initiation of ANC and associated factors among pregnant women attending antenatal care services in Agaro town health institutions, Southwest Ethiopia.

## Methods and materials

### Study design, area and period

Institution-based cross-sectional study design was conducted among pregnant women attending ANC services. The study was conducted from April 15 to June 15, 2019 in Agaro town, Southwest Ethiopia. Agaro town is located 390 km Southwest of Addis Ababa. The town has a population of 40,114 residents. Of these, 8,877 were women at the reproductive age (15–49 years). In the town, there were 8 governmental health facilities (two health centers, one governmental hospital and five health posts) and six private clinics. All health facilities that had well documented registration books and provided ANC services were included in the study. Of governmental health facilities, three of them (one hospital and two health centers) were included in the study. Of private health facilities, two private clinics were included in the study. The two-months report (January and February 2019) indicated that the number of pregnant women attending ANC services in the five eligible health facilities were 798 (338 from Agaro Hospital, 318 from Agaro health center, 54 from Razel health center, 42 from Yohannes clinic and 46 from Kegna clinic).

### Population

All pregnant women who were attending antenatal care services at health facilities of Agaro town during the data collection period were included in the study and pregnant women who were seriously sick and unable to communicate during the study period were excluded from the study.

### Sampling size and sampling procedures

The sample size was calculated using a single population proportion formula ($n = \frac{(z_{a/2})^2 * p(1-p)}{d^2}$) [16] based on the following assumptions: the proportion of timely initiation of ANC in the previous study, p = 35.4% [17], level of significance, α = 5%, 95% confidence interval and margin of error, d = 5%. Adding a 10% non-response rate, the minimum sample size was 387 pregnant women.

 All eligible health facilities were included in the study. The two-months report (January and February 2019) indicated that the number of pregnant women attending ANC services in the five eligible health facilities were 798 (338 from Agaro Hospital, 318 from Agaro health center, 54 from Razel health center, 42 from Yohannes clinic and 46 from Kegna clinic). Systematic random sampling technique was employed to recruit pregnant women. The total sample size (387) was proportionally allocated for the five health facilities depending on their client flow. The proportional allocation formula (ni = (n/N) *Ni) was used to determine the sample size for each health facility. Based on this, 164 pregnant women from Agaro hospital, 154 pregnant women from Agaro health center, 26 pregnant women from Razel health center, 21 pregnant women from Yohannes private clinic and 22 pregnant women from Kegna private clinic were recruited

to the study. The sampling interval ($K = N/n$) for each health facility was 2. The starting point was determined by balloting and the second pregnant woman attending the ANC service was the first client recruited for the study. Then, every other pregnant woman who was attending ANC services was recruited at each health facility until the total sample size for this study was filled.

## Variables

The dependent variable was timely initiation of an antenatal care visit. The independent variables were identified by reviewing different literatures related to the topic of interest [10–15]. The list of variables were socio-demographic characteristics of pregnant women such as age of the mother, marital status, occupation, educational status, place of residence, and monthly income. The other independent variables were previous use of ANC services, parity, knowledge about time of first ANC, distance from health facility, cost of services, previous birth outcome, medical complications in previous pregnancy, intention of pregnancy and satisfaction with ANC services of previous pregnancy.

## Data collection instruments and procedures

A structured and pre-tested face-to-face interviewer-administered closed-ended questionnaire was used to collect data. The questionnaire includes socio-demographic characteristics of pregnant women such as age of mother, marital status, occupation, educational status, residence, and monthly income. The other independent variables were previous use of ANC services, parity, knowledge on time of first ANC, distance from health facility, cost of services, previous birth outcome, medical complications in previous pregnancy, intention of pregnancy and satisfaction with ANC services of previous pregnancy [10–15]. Five trained female diploma nurses collected the data and the principal investigator and one supervisor supervised them. The data collectors were not employees of that institution.

The reliability of the questionnaire was checked using Cronbach's alpha reliability test and the value was 0.83. To ensure the quality of data, the questionnaire was prepared in English, translated to local language, and translated back to English to check for consistency. The data collectors and supervisors were trained for a day on data collection tools and procedures. Pre-test was done on 5% of the total sample size out of the study area prior to the time of actual data collection. Completeness and consistencies of the collected data were checked on a daily basis and feedback was given to data collectors based on findings.

## Data processing and analysis

The collected data was checked, cleaned and entered into Epi Data version 4.4.2 and was exported to and analyzed by SPSS version 25 software. Descriptive statistics such as frequency table, mean, median and standard deviation were used to summarize the data. The dependent variable was dichotomous, that is, timely (within 16 weeks of gestation) or late visit to ANC (after 16 weeks of gestation). The association between each independent variable and timely initiation of antenatal care was assessed using a bi-variable logistic regression model. Variables with p-value < 0.2 in the bi-variable logistic regression were included in the final model (multivariable logistic regression). In the final model, an adjusted odds ratio with a 95% confidence interval and a p-value < 0.05 was used to declare statistical significance.

## Ethical considerations

The study was conducted in accordance with the Declaration of Helsinki. Ethical clearance (CMHS 237/11/11) was obtained from the Institutional Review board of the College of

Medicine and Health Sciences, Wollo University. Permission letter to conduct the study was obtained from Agaro town health office. Written informed consent was obtained from pregnant women after they were informed about the objectives and procedures of the study. The right to refuse participation any time they want were assured. Any involvement in the study was done after complete consent was obtained. They were informed that all data obtained from them would be kept confidential by using codes instead of any personal identifiers and was meant only for the purpose of the study.

## Results

### Socio-demographic characteristics of pregnant women

Out of 387 pregnant women initially included in this study, 375 responded to the interview, which gave a response rate of 96.9%. About 292 (77.9%) women were in the age group of 20 to 34 years. The age group ranges from 16 to 42 years and the mean age of the respondents were 25.5 years (± 5.6 years). Regarding the ethnicity and religion of respondents, 243 (64.8%) and 254 (67.7%) were Oromo and Muslim, respectively. Three hundred eighteen (84.8%) ANC attendees were married. About 153 (40.8%) pregnant women have attended primary school and 191 (50.9%) pregnant women were housewives by occupation (**Table 1**).

### Obstetric history of pregnant women

About 284 (75%) pregnant women had more than one previous pregnancy experience. Fifty-eight (20.4%) pregnant women have had abortions. About 279 (74.4%) pregnant women had given one birth or above. Of these, 39 (14%) have experienced stillbirth. Thirty-six (12.9%) women had experienced pregnancy related complications. Of pregnant women who have had birth experience, 237 (84.9%) of the women have attended ANC services at various facilities (**Table 2**).

### Knowledge of pregnant women on timing of antenatal care

About 353 (94.1%) pregnant women reported that ANC service is important and 314 (83.7%) and 343 (91.5%) respondents knew that early booking of the first ANC improves the health of the mother and fetal outcome, respectively. Concerning the knowledge of appropriate time to begin first ANC, only 211 (56.3%) pregnant women reported that it should be within 16 weeks of the pregnancy. One hundred eighty (48%) pregnant women knew that pregnant women required four and above ANC visits (**Table 3**).

### Current pregnancy characteristics of pregnant women

Of 375 pregnant women, 214 (57.1%) have planned to be pregnant. Pregnancy conformed through urine test by checking the HCG hormone was found to be 133 (35.5%). About 226 (60.3%) women were advised to start ANC. One hundred fifty-one (40.3%) pregnant women reported that they experienced pregnancy related complications before they started the ANC service. Of those having experienced pregnancy related complications, 119 (78.8%) started their ANC follow up because of the complication (**Table 4**).

### Timing of first ANC visit of pregnant women

About 157 (41.9%) pregnant women started their first ANC visit timely. The timing of the first ANC booking ranged from 4 weeks to 40 weeks of gestation and the mean time of the ANC visit was 18 weeks (± 5.50 weeks).

**Table 1. Socio-demographic characteristics of pregnant women at Agaro town health institutions, Southwest Ethiopia, June 2019 (n = 375).**

| Variables | Number | Percentage |
|---|---|---|
| **Age** | | |
| 15–19 | 51 | 13.6 |
| 20–34 | 292 | 77.9 |
| 35+ | 32 | 8.5 |
| **Residence** | | |
| Rural | 154 | 41.1 |
| Urban | 221 | 58.9 |
| **Marital status** | | |
| Single | 23 | 6.1 |
| Married | 318 | 84.8 |
| Divorced | 23 | 6.1 |
| Widowed | 11 | 2.9 |
| **Religion** | | |
| Muslim | 254 | 67.7 |
| Orthodox Christian | 78 | 20.8 |
| Protestant | 37 | 9.9 |
| Others* | 6 | 1.6 |
| **Ethnicity** | | |
| Oromo | 243 | 64.8 |
| Amhara | 54 | 14.4 |
| Gurage | 31 | 8.3 |
| Silte | 22 | 5.9 |
| Others** | 25 | 6.6 |
| **Educational status** | | |
| No formal education | 82 | 21.9 |
| Primary school (1–8) | 153 | 40.8 |
| Secondary school (9–12) | 96 | 25.6 |
| Above secondary school | 44 | 11.7 |
| **Occupational status** | | |
| Government employee | 45 | 12.0 |
| Private worker | 108 | 28.8 |
| House wife | 191 | 50.9 |
| Student | 31 | 8.3 |
| **Family monthly income in ETB** | | |
| <1550 | 96 | 25.6 |
| 1550–5000 | 249 | 66.4 |
| >5000 | 30 | 8.0 |

**Others***: Waqefeta, Jehovah witness,

**Others****: Tigre, Wollaita, Kefa, Dawro

## Factors associated with timely initiation of antenatal care

A multivariable logistic regression was done to identify independent predictors of timely initiation of ANC. As a result, the analysis revealed that pregnant women who were primigravida, having good knowledge of timely initiation of ANC, planned to be pregnant and confirmed pregnancy by urine test were significantly associated with timely initiation of antenatal care.

**Table 2. Past obstetric history of pregnant women attending ANC services at Agaro town health institutions, Southwest Ethiopia, June 2019.**

| Variable | Number | Percentage |
|---|---|---|
| **Gravidity (n = 375)** | | |
| Primigravida | 91 | 24.3 |
| Multigravida | 284 | 75.7 |
| **History of abortion (n = 284)** | | |
| Yes | 58 | 20.4 |
| No | 226 | 79.6 |
| **Parity (n = 375)** | | |
| Nulliparous | 96 | 25.6 |
| Parity 1 and above | 279 | 74.4 |
| **History of still birth (n = 279)** | | |
| Yes | 39 | 14.0 |
| No | 240 | 86.0 |
| **Previous pregnancy complication (n = 279)** | | |
| Yes | 36 | 12.9 |
| No | 243 | 87.1 |
| **ANC visit for previous pregnancy (n = 279)** | | |
| Yes | 237 | 84.9 |
| No | 42 | 15.1 |
| **Advice when to start ANC for previous pregnancy (n = 237)** | | |
| Yes | 186 | 78.5 |
| No | 51 | 21.5 |
| **Satisfaction with ANC services for previous pregnancy (n = 237)** | | |
| Yes | 143 | 60.3 |
| No | 94 | 39.7 |

Pregnant women who were primigravida were 2.9 times more likely to initiate antenatal care timely as compared pregnant women who had pregnancy twice or more [AOR = 2.9, 95% CI: 1.6–5.3]. Pregnant women who had good knowledge of time of first ANC were about 3.8 times more likely to start first ANC visit timely as compared to their counterparts [AOR = 3.8, 95% CI: 2.2, 6.5]. Pregnant women with a planned pregnancy were 5.1 times more likely to start the first ANC visit timely as compared to those who had an unplanned pregnancy [AOR = 5.1, 95% CI: 2.9, 8.9]. Pregnant women who had confirmed their pregnancy by urine test were 4.1 times more likely to initiate ANC timely than those who confirmed pregnancy through a missed period. [AOR = 4.1, 95% CI = 2.4–6.9] (**Table 5**).

## Discussion

This study was aimed to determine the timely initiation of antenatal care and its associated factors among pregnant women attending antenatal care services in Agaro town health institutions, Southwest Ethiopia.

The percentage of timely initiation of ANC among pregnant women in the current study was consistent with studies conducted in Bahir Dar, Ethiopia and sub-Saharan Africa [18, 19]. However, the percentage was lower compared to a study done in Southeast Ethiopia [20]. The percentage was also lower than studies conducted in Nepal and Sindh, Pakistan [21, 22]. On the contrary, the percentage of timely initiation of ANC of the current study was higher compared to studies done in Dilla town [17] and Southern Ethiopia [12, 23]. 'The percentage was

**Table 3. Knowledge of pregnant women on timing of first ANC booking at Agaro Town health institutions, Southwest Ethiopia, June 2019.**

| Variables | Number | Percentage |
|---|---|---|
| **ANC is important** | | |
| Yes | 353 | 94.1 |
| No | 22 | 5.9 |
| **The right time to start ANC** | | |
| Within 16 weeks | 211 | 56.3 |
| After the 16 weeks | 164 | 43.7 |
| **Timing of ANC improves outcome of the fetus** | | |
| Yes | 314 | 83.7 |
| No | 61 | 16.3 |
| **Timing of ANC improves outcome of the mother** | | |
| Yes | 343 | 91.5 |
| No | 32 | 8.5 |
| **Frequency of ANC visit** | | |
| One | 15 | 4.0 |
| Two-three | 125 | 33.3 |
| Four-six | 180 | 48.0 |
| More than six | 55 | 14.7 |
| **All pregnant women are at risk of pregnancy complication** | | |
| Yes | 247 | 65.9 |
| No | 128 | 34.1 |
| **Difference between primiparous and multiparous in timing of first ANC visit** | | |
| Yes | 204 | 54.4 |
| No | 171 | 45.6 |

also higher compared to the finding of Tesfaye *et al* from a systematic review and meta-analysis of studies conducted in Ethiopia [7]. The possible reasons for this difference might be studies conducted in Southeast Ethiopia, Nepal and Sindh, Pakistan were on large sample size compared to the current study. In addition, the previous studies [7, 12, 14, 15, 17, 20–24] indicated that the difference in having previous experience of 1st ANC visit, getting advice about ANC visit, place of residence, cultures to reveal pregnancy at early time and educational status of the pregnant women might be the possible reasons.

Mothers who had no previous pregnancy experience were more likely to start first ANC visit timely compared to those with a gravidity of two and above. This finding was similar to a systematic review and meta-analysis study conducted in Ethiopia [7]. This might be because young women with first pregnancy and childbirth are more careful about pregnancy and, in turn, leads them to require antenatal care early compared to multigravida women. In addition, younger women might implement the advice they get from family members, neighbors, and health professionals more compared to the older ones. It might also be because multigravida women feel more confident [7] after previous experience and feel that they can manage the pregnancy process easily and there is no need to visit health facilities early.

This study revealed that pregnant women who had good knowledge of time of first ANC were more likely to start first ANC visit timely as compared to their counterparts. This study is in line with studies done in Mekele city, Ethiopia, and Ambo, Ethiopia [14, 15]. This might be because the more pregnant women know the time of first ANC visit, the more they know the advantages of commencing ANC visit. In turn, this might help them to start the ANC visit timely. Pregnant women who had an intention of getting pregnant were more likely to start

**Table 4. Current pregnancy characteristics for pregnant women attending ANC at Agaro town Health institutions, Southwest Ethiopia, June 2019.**

| Variables | Number | Percentage |
|---|---|---|
| **Intention of pregnancy (n = 375)** | | |
| Planned | 214 | 57.1 |
| Unplanned | 161 | 42.9 |
| **Pregnancy related complications for current pregnancy (n = 375)** | | |
| Yes | 151 | 40.3 |
| No | 224 | 59.7 |
| **ANC attendance due to complications (n = 151)** | | |
| **Yes** | 119 | 78.8 |
| **No** | 32 | 21.2 |
| **Means of conforming pregnancy (n = 375)** | | |
| Missed period | 242 | 64.5 |
| By urine test | 133 | 35.5 |
| **Paid for service (n = 375)** | | |
| Yes | 55 | 14.7 |
| No | 320 | 85.3 |
| **Advice when to start ANC (n = 375)** | | |
| Yes | 226 | 60.3 |
| No | 149 | 39.7 |
| **Source of advice (n = 226)** | | |
| Husband | 66 | 29.2 |
| Friend | 60 | 26.5 |
| Other family member | 37 | 16.4 |
| Media | 13 | 5.8 |
| Health professionals | 50 | 22.1 |

first ANC visit timely as compared to those who had an unplanned pregnancy. This finding was in line with a study done in Arbaminch, Ethiopia [25]. This might be due to planned pregnancies being more cared for by pregnant women and their spouses and, in turn, enables the pregnant women to book for ANC timely.

Pregnant women who had confirmed pregnancy by urine test were more likely to initiate ANC timely than those who confirmed their pregnancy through a missed period. This finding was consistent with a study conducted in Halaba [26]. This could be because urine is done in health institutions and pregnant women are initiated to start ANC when they come to confirm pregnancy.

Overall, the current study implies that efforts are needed to increase pregnant women's knowledge of antenatal care and its right time of commencement by providing information for pregnant women about the importance of utilizing timely ANC, pregnancy risks, and danger signs. Including both public and private health facilities to reflect the real timely initiation and associated factors of ANC in the study area was the strength of the study. The cross-sectional nature of the study could not allow us to make strong cause-effect associations and the generalizability of the study to a larger population might be limited since it was conducted at health institutions. Since the data were collected from an interviewer-administered questionnaire, there might be a possibility of social desirability bias.

## Conclusions

Despite the efforts made to make ANC visit services freely available, timely initiation of antenatal care among pregnant women in the study area was low. Pregnant women who were

**Table 5. Factors associated with timely initiation of ANC among pregnant women at Agaro town health institution, Southwest Ethiopia, June 2019.**

| Variables | Timely ANC initiation (No) | Late ANC Initiation (No) | COR 95% CI | AOR 95% CI |
|---|---|---|---|---|
| **Knowledge on ANC** | | | | |
| Yes | 137 | 118 | 4.7 [3.0–7.5] | 3.8 [2.2, 6.5] |
| No | 20 | 100 | 1 | 1 |
| **Gravidity** | | | | |
| Primigravida | 60 | 31 | 3.7 [2.3–6.1] | 2.6 [1.6, 5.3] |
| Multigravida | 97 | 187 | 1 | 1 |
| **Intention of pregnancy** | | | | |
| Planned | 123 | 95 | 4.9 [3.1–7.8] | 5.1 [2.9, 8.9] |
| Unplanned | 33 | 124 | 1 | 1 |
| **Pregnancy checking method** | | | | |
| By menstruation | 68 | 174 | 1 | 1 |
| By using urine test | 89 | 44 | 5.2 [3.3–8.2] | 4.1 [2.4, 6.9] |

**Keys:** ANC: Antenatal care; AOR: Adjusted odds ratio; CI: Confidence interval; COR: Crude odds ratio

primigravida, having good knowledge of timely initiation of ANC, planned to be pregnant, and confirmed pregnancy by urine test were significantly associated with timely initiation of antenatal care. Therefore, efforts that strengthen and maintain local information dissemination networks on antenatal care and its right time of commencement, increase pregnant women's knowledge of the timing of ANC service by providing information about the importance of utilizing timely ANC, pregnancy risks, and danger signs and reducing unplanned pregnancies should be organized and implemented.

## Supporting information

**S1 File.**
(RAR)

**S1 Data.**
(SAV)

## Acknowledgments

Our thanks go to all pregnant women for their participation. Our deepest gratitude also goes to Wollo University, data collectors, and Agaro town health office.

## Author Contributions

**Conceptualization:** Toffik Redi, Getaw Walle Bazie.

**Data curation:** Toffik Redi, Oumer Seid, Getaw Walle Bazie.

**Formal analysis:** Toffik Redi, Oumer Seid, Getaw Walle Bazie, Erkihun Tadesse Amsalu, Niguss Cherie, Melaku Yalew.

**Investigation:** Toffik Redi, Niguss Cherie.

**Methodology:** Toffik Redi, Getaw Walle Bazie, Erkihun Tadesse Amsalu, Niguss Cherie, Melaku Yalew.

**Project administration:** Toffik Redi, Niguss Cherie.

**Resources:** Niguss Cherie.

**Software:** Toffik Redi, Oumer Seid, Getaw Walle Bazie, Erkihun Tadesse Amsalu, Melaku Yalew.

**Supervision:** Melaku Yalew.

**Writing – original draft:** Toffik Redi, Oumer Seid, Getaw Walle Bazie.

**Writing – review & editing:** Oumer Seid, Getaw Walle Bazie, Erkihun Tadesse Amsalu, Melaku Yalew.

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
