## [Decision Letter · Decision Letter 0]

12 Jan 2022

PONE-D-21-21466

Timely Initiation of Antenatal Care and Associated Factors among Pregnant Women Attending Antenatal Care in Southwest Ethiopia

PLOS ONE

Dear Dr. Bazie,

Thank you for submitting your manuscript to PLOS ONE. After careful consideration, we feel that it has merit but does not fully meet PLOS ONE’s publication criteria as it currently stands. Therefore, we invite you to submit a revised version of the manuscript that addresses the points raised during the review process.

The reviewers have recommended that the manuscript be rewritten with particular attention given to clearly explaining the rationale, methods, and conclusions.

We look forward to receiving your revised manuscript.

Kind regards,

Nancy Beam, PhD

Academic Editor

PLOS ONE

Journal Requirements:

6. We noticed you have some minor occurrence of overlapping text with the following previous publication(s), which needs to be addressed:

- http://www.bioline.org.br/request?rh13063

In your revision ensure you cite all your sources (including your own works), and quote or rephrase any duplicated text outside the methods section. Further consideration is dependent on these concerns being addressed.

Reviewers' comments:

Reviewer's Responses to Questions

**Comments to the Author**

1. Is the manuscript technically sound, and do the data support the conclusions?

Reviewer #1: Yes

Reviewer #2: Partly

Reviewer #3: Yes

Reviewer #4: Partly

Reviewer #5: Yes

2. Has the statistical analysis been performed appropriately and rigorously? 

Reviewer #1: I Don't Know

Reviewer #2: Yes

Reviewer #3: No

Reviewer #4: Yes

Reviewer #5: Yes

3. Have the authors made all data underlying the findings in their manuscript fully available?

Reviewer #1: Yes

Reviewer #2: Yes

Reviewer #3: Yes

Reviewer #4: Yes

Reviewer #5: Yes

4. Is the manuscript presented in an intelligible fashion and written in standard English?

Reviewer #1: Yes

Reviewer #2: No

Reviewer #3: Yes

Reviewer #4: No

Reviewer #5: Yes

5. Review Comments to the Author

Reviewer #1: Thank you for giving me the opportunity to review this paper.

My comments are the following:

Background

The background should be written at a higher level as we can assume that readers have basic understanding of the topic. For example, ANC does not need to be defined.

The background should try to make a better link between timely initiation of ANC and maternal mortality

I would suggest using local or regional references when discussing factors influencing timely initiation ofANC. The references are also quite outdated and would recommend using more current references such as:

https://bmcpregnancychildbirth.biomedcentral.com/articles/10.1186/s12884-020-03236-9

https://www.ncbi.nlm.nih.gov/pmc/articles/PMC7864666/

Methods

The methods should be re-written for better flow

The section of population is very repetitive and can be summarized in one sentence

Is Agari a rural, peri-urban or urban setting?

How did you choose the independent variables? Please include a brief discussion or some references.

A section on operational definition is not needed and can be incorporated earlier in the methods section

How many centres/ institutions were eligible? How did you choose the five centres? Did you expect to have differences between public and private institutions - is that why you selected a mix of both?

The section on Ethical Considerations should be shortened

Results

Please re-visit the titles of the different sections of the results. I would suggest perhaps consolidating some of these.

Discussion

Please structure the discussion to include strengths, limitations and policy implications

In the discussion, the authors make mention several times that the study findings are in keeping with findings of studies in other places in Ethiopia. Could you elaborate on whether these other places are similar to your study setting?

Please review the paper for English language/ grammar.

Reviewer #2: This descriptive study highlights an important issue in the unique setting of Town Agaro town in Ethiopia with potentials to be basis of actionable plans towards getting pregnant women to attend ANC in timely manner.

However the issues that should be addressed are

1. Impact of untimely ANC attendance was not well captured- maternal mortality was not linked to the issue under study; thus and also the rationale for the study in SW Ethiopia was unclear.

2. The Setting of study is not easily understood eg- a. what type of health institutions exist in Agaro from which the facilities used were drawn from?-private with fee for service , primary, secondary, etc? it should be described in setting

b. How did the authors selected the ones used.

c. What is meant by rural and urban areas in the same town as used as a variables

d. source of data/figures- such as a. number of pregnant women in the town.b. the two month report of 798 of which institution and whose data?

3.. Population- the term seriously sick as an exclusive criteria is subjective- does it mean mothers who were hospitalized?

4. Variables- Many Variables were not clearly defined-

a. is being a House wife synonymous with being economically unemployed?

b.,waiting time( where?-before seeing the health worker or collecting lab results?),

c. cost of services-( confusing as the background pointed to free services in Agora health facilities).

d. The cash / money classification in ETB means little if not explained, standardized or used in a socioeconomic classification.

e. satisfaction with services- How can a future event determine the timeliness of ANC, in other words satisfaction cant be assessed from the point at when they were yet to use the ANC services they were yet to use the ANC services; except it meant satisfaction with previous use

5. Ethics- while the single pregnant under 18yrs could have parental permission , a married minor is not a legal minor she may give consent by herself .

6. Results- within the tables, variables should be stated not as questions

7. The use of language - the word Lottery, i guess meant balloting, 'intention of pregnancy' should be planned or unplanned pregnancy.

8. Results should be discussed not re-stated in discussions. comparisons of trends or percentages with previous studies should be clear with possible reasons truely related to findings

Reviewer #3: Abstract

Methods: Line 31-32: The authors should revised the statement "Structured and pretested face-to-face

32 interview technique was used to collect data"

The authors should add the sampling method to the methods section.

Background

Lines 58-59: WHO has provided an updated recommended number of ANC visits and the authors should replace it with what they have in their manuscript currently.

Lines 65-67: The sentences in these lines lack adequate citations.

Line 68 and Line 72: its sub-Saharan Africa not Sub-Saharan Africa

Methods

The authors should provide more detailed description of the study setting highlighting the health facilities and if possible the interventions put in place to improve maternal health service utilization.

Sampling size and sampling procedure

The authors should provide the formula used to calculate the sample size and support it with the appropriate reference.

The authors should categorically state the sampling method employed in the and clearly describe how it was used to recruit the pregnant women.

Data collection

How did the authors check for the reliability and validity of the questionnaire?

Analysis

Did the authors apply weighting in the analysis?

Ethical consideration

The authors should provide the ethical clearance number.

Results

Table 3 and Table 4: The authors should make sure that all the variables that look like questions have been revised to look like variables.

Line 200 and 211: The headings in these lines should be changed to sentence case.

Why did the authors not perform regression analysis for the variables in Table 1 and Table 2. Any special reason for not doing so??

I suggest the authors reanalyze the data to include those variables in the regression analysis.

Reviewer #4: I appreciate the work of the authors on identifying the factors associated with timely initiation of antenatal care in southwest Ethiopia. The authors highlight a major public health concern with widespread implications in an understudied population. Conclusions are both valid and meaningful. However, I have highlighted below some significant analytical concerns that should be addressed in the paper:

MAJOR COMMENTS

The manuscript would benefit from heavy editorializing. The grammar can be strengthened to avoid detraction from the value of the study.

INTRODUCTION

-I appreciate that the authors keep the introduction short and succinct.

METHODS

-The description of the sampling method is a bit unclear. The authors write that the proportion of timely initiation of ANC is 35.4%. Was this number calculated from the present study, or from the cited 2014 manuscript? What is the rationale for assuming there is a 10% non-response rate? Could the authors please expand upon the lottery method mentioned in lines 122-123?

-The authors would benefit from further discussion of the format of the questionnaire used to collect data. Were questions open-ended, agree/disagree, yes/no, etc.?

RESULTS

-The authors write that 20% of women with prior pregnancies had an abortion. Were these all results of elective procedures, or does this figure include miscarriages?

-How did mothers know they were experiencing complications (UTIs/high blood pressure) before their first ANC visit? Did the first visit encourage them to seek subsequent care?

DISCUSSION

-What is the rationale for comparing southwest Ethiopia to Nepal and Pakistan? The nations and cultures are entirely different, and although all three countries are low-to-middle income regions, it doesn't make sense to compare them directly in this context.

-Although it is logical, are there any sources that can support the claim that women who have other children feel more confident and therefore are less likely to seek timely care? If not, please mention that formal analyses have not confirmed the authors' suspicions.

CONCLUSION

-The manuscript would benefit from a brief expansion of recommended public health practices.

Reviewer #5: The introduction has set a good context for the study and methods are sound and clearly outline the procedures. This is a well written paper. However a few comments were observed below.

Introduction

There is a paper quoted here: ''WHO Geneva. Carla A, Tessa W, Blanc A, Van P, et al. ANC in developing countries, 352 promises, achievements, and missed opportunities; an analysis of trends, levels and 353 Differentials, 1990-2001. 2003''. This research from 2003 is quite old. It would be beneficial to more recent evidence to give us a better picture and trend of timing of ANC visits.

Discussion

Expatiate on efforts to be made to increase pregnant women’s knowledge on ANC and the right time for commencement, to include some practical examples.

Why was the more recent EHDS from 2019 not used? The maternal health care chapter contains information ANC timing.

6. PLOS authors have the option to publish the peer review history of their article (what does this mean?). If published, this will include your full peer review and any attached files.

Reviewer #1: No

Reviewer #2: No

Reviewer #3: No

Reviewer #4: No

Reviewer #5: **Yes: **Fatima Abdulaziz Sule

---

## [Author Response · Author response to Decision Letter 0]

21 Feb 2022

Response to Reviewers and Editor 

Firstly, I would like to thank editor and reviewers for your valuable and constructive comments and questions that are helpful for improving the quality of the manuscript. The responses for each point are done as follows.

To Editor 

1. The manuscript is done based on the journal requirements

2. The questionnaire as a supporting information is separately attached

3. The data availability statement is described within the manuscript

4. Regarding providing repository information, we have changed our mind. We will provide the necessary information upon request by the journal.

5. Ethics statement out of the methods section is deleted 

6. Regarding some minor occurrence of overlapping text with the following publication (http://www.bioline.org.br/request?rh13063), we checked it and by now the whole manuscript is revised to avoid any overlap. 

Response to Reviewers

Reviewer #1: Thank you for giving me the opportunity to review this paper.

My comments are the following:

Background

• The background should be written at a higher level as we can assume that readers have basic understanding of the topic. For example, ANC does not need to be defined.

Response: The whole background section is revised and necessary changes are made to make it at higher level and the definition of ANC is removed.

• The background should try to make a better link between timely initiation of ANC and maternal mortality

Response: A statement that describes the link between timely initiation of ANC and maternal mortality is added in the revised manuscript (please see paragraphs 3 & 4 of the current version)

• I would suggest using local or regional references when discussing factors influencing timely initiation of ANC. The references are also quite outdated and would recommend using more current references such as:

https://bmcpregnancychildbirth.biomedcentral.com/articles/10.1186/s12884-020-03236-9

https://www.ncbi.nlm.nih.gov/pmc/articles/PMC7864666/

Response: Local and regional references are used to discuss the factors associated with timely initiation of ANC (Please see references 10-12) and all outdated references are replaced by the current references including the references you suggested us. (Please see references 1-15)

Methods

• The methods should be re-written for better flow

Response: Correction is made throughout the methods section to make the flow better. 

• The section of population is very repetitive and can be summarized in one sentence

Response: Summarized in one sentence and the repetition is corrected (Please see population section of the revised manuscript) 

• Is Agaro a rural, peri-urban or urban setting?

Response: Agaro is an urban setting. However, clients are coming from both the town and the surrounding rural areas to get ANC services. 

• How did you choose the independent variables? Please include a brief discussion or some references.

Response: The description and the references are included (Please see the variables section of the revised manuscript) 

• A section on operational definition is not needed and can be incorporated earlier in the methods section

Response: Deleted as per your suggestion.

• How many centres/ institutions were eligible? How did you choose the five centres? Did you expect to have differences between public and private institutions - is that why you selected a mix of both?

Response: In the town, there were two health centers, one governmental hospital, six private clinics and five health posts. Of these, five health facilities (three government health institutions (one hospital and two health centers) and two private clinics) that had well documented registration books and providing ANC services were eligible and included in the study. (Please see study design, area and period section). Yes, we expect differences between public and private institutions and mostly researchers are focusing on public facilities only. So, to ensure the external validity of the findings to similar settings both public and private facilities were included. 

• The section on Ethical Considerations should be shortened

Response: Shortened as per your comment (Please see Ethical Considerations section of the revised manuscript)

Results

• Please re-visit the titles of the different sections of the results. I would suggest perhaps consolidating some of these.

Response: Revisit is done on the titles of the different section of the results. However, consolidating the titles was not made because each title discusses different issues and have its own table. 

Discussion

• Please structure the discussion to include strengths, limitations and policy implications

Response: Strengths, limitations and policy implications are included (Please see the last paragraph of the discussion section)

In the discussion, the authors make mention several times that the study findings are in keeping with findings of studies in other places in Ethiopia. Could you elaborate on whether these other places are similar to your study setting?

Response: Some of the studies are conducted in Southern Ethiopia relatively nearest to the current study area. Some others are conducted in Northwest and Eastern Ethiopia. Overall, most of the areas are similar to the study setting. Some others are different. The health service provision culture in Ethiopia is more or less similar. However, there might be variation in terms of providing 1st ANC visit and the factors associated to it from area to area within the country. That’s why the current study is conducted to appreciate that difference and design appropriate area specific interventions. 

• Please review the paper for English language/ grammar.

Response: The whole manuscript is revised to correct grammatical and writing errors.

Reviewer #2: This descriptive study highlights an important issue in the unique setting of Town Agaro town in Ethiopia with potentials to be basis of actionable plans towards getting pregnant women to attend ANC in timely manner.

However, the issues that should be addressed are:

1. Impact of untimely ANC attendance was not well captured- maternal mortality was not linked to the issue under study; thus, and also the rationale for the study in SW Ethiopia was unclear.

Response: Impact of untimely initiation of ANC is discussed and linked to maternal mortality (please see paragraph 4 of the current version) and the rationale for the study is revised (please see the last paragraph of the background section) 

2. The setting of study is not easily understood eg- a. what type of health institutions exist in Agaro from which the facilities used were drawn from? private with fee for service, primary, secondary, etc? it should be described in setting

Response: In the town, there were two health centers, one governmental hospital, six private clinics and five health posts. Of these, three government health institutions (one hospital and two health centers) and two private clinics that had well documented registration books and providing ANC services were eligible and included in the study. (Please see study design, area and period section).

b. How did the authors selected the ones used.

Response: All health facilities that had well documented registration books and were providing ANC services were eligible and included in the study. Then, the sample size was proportionally allocated to each eligible health facility as per their client flow. 

c. What is meant by rural and urban areas in the same town as used as a variables

Response: Though the health facilities are found in the town, the clients were coming from both urban area (Agaro town) and rural areas surrounding the town to get ANC services. That is why, the variable place of residence was used. 

d. source of data/figures- such as a. number of pregnant women in the town. b. the two-month report of 798 of which institution and whose data?

Response: The detail description is made (Please see study design, area and period section).

3.. Population- the term seriously sick as an exclusive criterion is subjective- does it mean mothers who were hospitalized?

Response: It doesn’t mean hospitalized mothers. What we excluded were pregnant women who came to the health facilities for ANC service but unable to make the interview due to their sickness. 

3. Variables- Many Variables were not clearly defined-

a. is being a House wife synonymous with being economically unemployed?

Response: Being a house wife in our context is if a pregnant woman is engaged in raising children and taking care of her family and the family income is based on husband’s income. So, housewives are economically unemployed. 

b. waiting time (where? -before seeing the health worker or collecting lab results?),

Response: Waiting time is deleted from the variables list since it is not discussed in the results section. 

c. cost of services- (confusing as the background pointed to free services in Agaro health facilities).

Response: Since private health facilities were included in the study cost of service (paid for service) is included in the variables list. 

d. The cash/money classification in ETB means little if not explained, standardized or used in a socioeconomic classification.

Response: It was done based on previous studies for sake of comparison. However, it was not found to be significant in the current study. 

e. satisfaction with services- How can a future event determine the timeliness of ANC, in other words satisfaction can’t be assessed from the point at when they were yet to use the ANC services, they were yet to use the ANC services; except it meant satisfaction with previous use

Response: That was writing error. Corrected as ‘satisfaction with ANC services for previous pregnancy’.

4. Ethics- while the single pregnant under 18yrs could have parental permission, a married minor is not a legal minor she may give consent by herself.

Response: Thank you for the lesson you thought us. we removed that part. (Please see Ethical Considerations section of the revised manuscript)

5. Results- within the tables, variables should be stated not as questions

Response: All variables stated in question form are changed to statement form (Please see tables 3 and 4)

6. The use of language - the word Lottery, i guess meant balloting, 'intention of pregnancy' should be planned or unplanned pregnancy.

Response: The issue of lottery method is described in the current version of sample size and sampling procedures section. (Please see sample size and sampling procedures section of the current version). The variable ‘pregnancy planned’ is changed as ‘intention of pregnancy’ and the categories are changed as ‘planned pregnancy and unplanned pregnancy’. (Please see table 4)

7. Results should be discussed not re-stated in discussions. comparisons of trends or percentages with previous studies should be clear with possible reasons truly related to findings

Response: The problem of re-stating is corrected and we tried to clarify the possible reasons for comparisons of trends or percentages with the previous studies. 

Reviewer #3: 

Abstract

• Methods: Line 31-32: The authors should revise the statement "Structured and pretested face-to-face interview technique was used to collect data"

Response: Changed as “A structured and pre-tested face-to-face interviewer-administered questionnaire was used to collect data.”

• The authors should add the sampling method to the methods section.

Response: Added as “Systematic random sampling technique was employed to recruit pregnant women.”

Background

• Lines 58-59: WHO has provided an updated recommended number of ANC visits and the authors should replace it with what they have in their manuscript currently.

Response: Replaced (please see paragraph 3 of the current version)

• Lines 65-67: The sentences in these lines lack adequate citations.

Response: Citations are added (please see paragraph 1 of the current version)

• Line 68 and Line 72: its sub-Saharan Africa not Sub-Saharan Africa

Response: Corrected (please see paragraphs 1 and 5 of the current version)

Methods

• The authors should provide more detailed description of the study setting highlighting the health facilities and if possible, the interventions put in place to improve maternal health service utilization.

Response: The detail description of the study setting is made (Please see study design, area and period section).

Sampling size and sampling procedure

• The authors should provide the formula used to calculate the sample size and support it with the appropriate reference.

Response: The formula is provided with appropriate reference (Please see reference 16)

• The authors should categorically state the sampling method employed in the and clearly describe how it was used to recruit the pregnant women.

Response: Brief description is made in the current version of sample size and sampling procedures section. 

Data collection

• How did the authors check for the reliability and validity of the questionnaire?

Response: We checked the reliability of the questionnaire using Cronbach’s alpha reliability test and the value was 0.83. The validity of the questionnaire was checked by conducting pre-test, training supervisors and data collectors and translating the questionnaire to the local language and translating back to English. (Please see Data collection instruments and procedures section of the current version)

Analysis

• Did the authors apply weighting in the analysis?

Response: We didn’t apply weighting in the analysis. 

Ethical consideration

• The authors should provide the ethical clearance number.

Response: We put the ethical clearance number (Please see Ethical Considerations section of the revised manuscript)

Results

• Table 3 and Table 4: The authors should make sure that all the variables that look like questions have been revised to look like variables.

Response: All variables stated in question form are changed to statement form (Please see tables 3 and 4)

• Line 200 and 211: The headings in these lines should be changed to sentence case.

Response: The headings are changed to sentence case. 

• Why did the authors not perform regression analysis for the variables in Table 1 and Table 2? Any special reason for not doing so?? I suggest the authors reanalyze the data to include those variables in the regression analysis.

Response: We did regression analysis for all the variables included in the study including those listed in Table 1 and Table 2. However, only four variables (including gravidity from table 2) in table 5 were found to be significantly associated with timely initiation of antenatal care. 

Reviewer #4: 

I appreciate the work of the authors on identifying the factors associated with timely initiation of antenatal care in southwest Ethiopia. The authors highlight a major public health concern with widespread implications in an understudied population. Conclusions are both valid and meaningful. However, I have highlighted below some significant analytical concerns that should be addressed in the paper:

MAJOR COMMENTS

The manuscript would benefit from heavy editorializing. The grammar can be strengthened to avoid detraction from the value of the study.

Response: The whole manuscript is revised to correct grammatical and writing errors. 

INTRODUCTION

• I appreciate that the authors keep the introduction short and succinct.

Response: We tried to make it short and succinct by not missing the necessary information that should be included. 

METHODS

• The description of the sampling method is a bit unclear. The authors write that the proportion of timely initiation of ANC is 35.4%. Was this number calculated from the present study, or from the cited 2014 manuscript? What is the rationale for assuming there is a 10% non-response rate? Could the authors please expand upon the lottery method mentioned in lines 122-123?

Response: We tried to describe the sampling method clearly. The proportion of timely initiation of ANC, 35.5% was taken from the cited 2014 paper to estimate the sample size of the current study. The rationale for taking 10% non-response rate was to compensate invalid filled questionnaires due to incompleteness and writing errors. The way the lottery method used to recruit the first pregnant women (starting point) is more briefed in the current version of the manuscript. 

• The authors would benefit from further discussion of the format of the questionnaire used to collect data. Were questions open-ended, agree/disagree, yes/no, etc.?

Response: The questions included in the questionnaire were closed-ended (Please see Data collection instruments and procedures section of the current version)

RESULTS

• The authors write that 20% of women with prior pregnancies had an abortion. Were these all results of elective procedures, or does this figure include miscarriages?

Response: It includes both elective procedures and miscarriages based on the report we got from pregnant women through the interviewer-administered questionnaire. 

• How did mothers know they were experiencing complications (UTIs/high blood pressure) before their first ANC visit? Did the first visit encourage them to seek subsequent care?

Response: It is talking about the complications for the previous pregnancy. When they attended their previous ANC, they knew their overall complication status. 

DISCUSSION

• What is the rationale for comparing southwest Ethiopia to Nepal and Pakistan? The nations and cultures are entirely different, and although all three countries are low-to-middle income regions, it doesn't make sense to compare them directly in this context.

Response: Primarily, we tried to compare the study findings to the previous findings in other places of Ethiopia. To expand the comparison outside Ethiopia, we selected countries like Nepal, Pakistan and so on from developing countries. So, the rationale is expanding comparison of the findings to other developing countries assuming that they are similar in terms of health service utilization including ANC visit. 

• Although it is logical, are there any sources that can support the claim that women who have other children feel more confident and therefore are less likely to seek timely care? If not, please mention that formal analyses have not confirmed the authors' suspicions.

Response: The explanation was based on evidence (Please see reference 25).

CONCLUSION

• The manuscript would benefit from a brief expansion of recommended public health practices.

Response: Modification is made 

Reviewer #5: The introduction has set a good context for the study and methods are sound and clearly outline the procedures. This is a well written paper. However, a few comments were observed below.

Introduction

• There is a paper quoted here: ''WHO Geneva. Carla A, Tessa W, Blanc A, Van P, et al. ANC in developing countries, 352 promises, achievements, and missed opportunities; an analysis of trends, levels and 353 Differentials, 1990-2001. 2003''. This research from 2003 is quite old. It would be beneficial to more recent evidence to give us a better picture and trend of timing of ANC visits.

Response: Thank you really. All old references are removed and replaced by the recent ones (please see references 1-15 of the current version)

Discussion

• Expatiate on efforts to be made to increase pregnant women’s knowledge on ANC and the right time for commencement, to include some practical examples.

Response: Modification is done with some practical examples. (Please see the last paragraph of the discussion section) 

• Why was the more recent EHDS from 2019 not used? The maternal health care chapter contains information ANC timing.

Response: We used EDHS 2016 report to cite the information ‘only 20% of women had their first ANC during the first trimester’. However, the mini-EDHS 2019 doesn’t have information regarding the percentage of women who had their first ANC during the first trimester. When the full 2019 EDHS is released, it might be included.

---

## [Decision Letter · Decision Letter 1]

26 Apr 2022

PONE-D-21-21466R1Timely Initiation of Antenatal Care and Associated Factors among Pregnant Women Attending Antenatal Care in Southwest EthiopiaPLOS ONE

Dear Dr. Bazie,

Thank you for submitting your manuscript to PLOS ONE. After careful consideration, we feel that it has merit but does not fully meet PLOS ONE’s publication criteria as it currently stands. Therefore, we invite you to submit a revised version of the manuscript that addresses the points raised during the review process.

Please address the points raised by Reviewer #2 below. In particular, please ensure you respond fully to their comments regarding providing explanation, discussion, or citations for statements made in the discussion of findings (point 4).

We look forward to receiving your revised manuscript.

Kind regards,

Hugh Cowley

Staff Editor

PLOS ONE

Reviewers' comments:

Reviewer's Responses to Questions

**Comments to the Author**

1. If the authors have adequately addressed your comments raised in a previous round of review and you feel that this manuscript is now acceptable for publication, you may indicate that here to bypass the “Comments to the Author” section, enter your conflict of interest statement in the “Confidential to Editor” section, and submit your "Accept" recommendation.

Reviewer #2: (No Response)

Reviewer #3: All comments have been addressed

Reviewer #5: All comments have been addressed

2. Is the manuscript technically sound, and do the data support the conclusions?

Reviewer #2: Yes

Reviewer #3: Yes

Reviewer #5: Yes

3. Has the statistical analysis been performed appropriately and rigorously? 

Reviewer #2: Yes

Reviewer #3: Yes

Reviewer #5: Yes

4. Have the authors made all data underlying the findings in their manuscript fully available?

Reviewer #2: Yes

Reviewer #3: Yes

Reviewer #5: Yes

5. Is the manuscript presented in an intelligible fashion and written in standard English?

Reviewer #2: No

Reviewer #3: Yes

Reviewer #5: Yes

6. Review Comments to the Author

Reviewer #2: This study has been conducted rigorously, with large enough sample size to represent the population of interest.The data supports the overall conclusion of the authors, however the major areas the authours need to look into is the language - words or sentences used caused difficulty in comprehension

1. the authors may be unwittingly passing the message that WHO has a 'recommended proportion of women' that should have timely ANC visit from the statements in lines 44, 77, 78- it should be re-worded .

2- in the Result section

i. variables should not be stated as questions

ii. line 198- 'Times of ANC visit' rather - ' frequency of ANC visits'

iii. Line 209 Does this complication made you start ANC rather -' ANC attendance due to complications' [ should also not be in question format ]

3. in line 243-247- in discussing and comparing the proportion /percentage of women with timely ANC visit , the reseachers repeatedly mention the phrase ' the finding was lower/ much higher' - it should be stated that it is percentage which is being discussed.

4. In proferring possible underlying in discussion of findings - many were stating random reasons with little scientific or factual evidence or citations eg- line 248 possible reasons for differences between current finding and other studies, the authors gave a string of reasons - " awareness, service coverage committement and training of healthworkers " but no explanation, discussion or citations

Recommended that the authors get help in re-writing for sense and flow before resubmission. Also some more effort into providing contextual and logical hypothesis /premise for various findings which would be beneficial to the authors and reader alike.

Reviewer #3: (No Response)

Reviewer #5: (No Response)

7. PLOS authors have the option to publish the peer review history of their article (what does this mean?). If published, this will include your full peer review and any attached files.

Reviewer #2: No

Reviewer #3: No

Reviewer #5: No

---

## [Author Response · Author response to Decision Letter 1]

3 Jun 2022

Response to Reviewers and Editor 

Firstly, we would like to thank the academic editor and reviewers for your additional valuable and constructive comments and questions that are helpful for improving the quality of the manuscript. The responses for each point are as follows.

To Academic Editor 

• The points raised by reviewer #2 are addressed.

• A rebuttal letter, Revised Manuscript with Track Changes and Manuscript without track changes are prepared. 

Response to Reviewers

Reviewer #2: This study has been conducted rigorously, with a large enough sample size to represent the population of interest. The data supports the overall conclusion of the authors, however the major areas the authors need to investigate are the language - words or sentences used caused difficulty in comprehension

1. the authors may be unwittingly passing the message that WHO has a 'recommended proportion of women' that should have timely ANC visit from the statements in lines 44, 77, 78- it should be re-worded.

Response: Thank you for your critical comment. Revision is done (Please see the conclusion part of the abstract and the conclusion section in the main document)

2. in the Result section

i. variables should not be stated as questions

Response: Corrected (Please see table 2)

ii. line 198- 'Times of ANC visit' rather - ' frequency of ANC visits'

Response: Corrected (Please see Table 3)

iii. Line 209 Does this complication made you start ANC rather -' ANC attendance due to complications' [should also not be in question format]

Response: Corrected (Please see Table 4)

3. in line 243-247- in discussing and comparing the proportion /percentage of women with timely ANC visit, the researchers repeatedly mention the phrase ' the finding was lower/ much higher' - it should be stated that it is percentage which is being discussed.

Response: Corrected (Please see the second paragraph of the discussion section) 

4. In proffering possible underlying in discussion of findings - many were stating random reasons with little scientific or factual evidence or citations e.g.- line 248 reasons for differences between current finding and other studies, the authors gave a string of reasons - " awareness, service coverage commitment and training of health workers " but no explanation, discussion, or citations

Response: Corrected (Please see the second paragraph of the discussion section)

---

## [Decision Letter · Decision Letter 2]

19 Jul 2022

PONE-D-21-21466R2Timely Initiation of Antenatal Care and Associated Factors among Pregnant Women Attending Antenatal Care in Southwest EthiopiaPLOS ONE

Dear Dr. Bazie,

Thank you for submitting your manuscript to PLOS ONE. After careful consideration, we feel that it has merit but does not fully meet PLOS ONE’s publication criteria as it currently stands. Therefore, we invite you to submit a revised version of the manuscript that addresses the points raised during the review process. Please address Reviewer 2's outstanding concerns regarding clarifications in the reporting, methodology justification, and language/copyediting.

We look forward to receiving your revised manuscript.

Kind regards,

Avanti Dey, PhD

Staff Editor

PLOS ONE

Reviewers' comments:

Reviewer's Responses to Questions

**Comments to the Author**

1. If the authors have adequately addressed your comments raised in a previous round of review and you feel that this manuscript is now acceptable for publication, you may indicate that here to bypass the “Comments to the Author” section, enter your conflict of interest statement in the “Confidential to Editor” section, and submit your "Accept" recommendation.

Reviewer #2: (No Response)

Reviewer #3: All comments have been addressed

2. Is the manuscript technically sound, and do the data support the conclusions?

Reviewer #2: Yes

Reviewer #3: Yes

3. Has the statistical analysis been performed appropriately and rigorously? 

Reviewer #2: Yes

Reviewer #3: Yes

4. Have the authors made all data underlying the findings in their manuscript fully available?

Reviewer #2: Yes

Reviewer #3: Yes

5. Is the manuscript presented in an intelligible fashion and written in standard English?

Reviewer #2: No

Reviewer #3: Yes

6. Review Comments to the Author

Reviewer #2: A. While the authors have attempted to address some of the comments , they failed to work on the general manuscript to improve its intelligibility in standard English as preiuosly advised - this is the major weaknes of this manuscript. They may do well to employ the services of an editor to help with the written language.

B.

Explanations for findings are not satisfactory mostly differences were just generic, not much specific explanations of differences between thier findings and the cited works - eg in line 261-266 " The possible reasons for this difference might be due to the difference in accessing information regarding timely initiation of antenatal care visit, advocating the importance of timely initiation of antenatal care, and advising pregnant women on the early booking of ANC, educational status of the pregnant women (14-15, 26). In addition, methodological variations among studies and the difference socio-demographic characteristics of the study participants might be the possible reasons-

C... specific isssues that should have been worked on and still noted are stated below

1. line 27- should be- late in pregnancy , not in their late pregnancy; check and reword line47; line 50-60

2.line 87- the statement -"That is lower than the WHO standard (14, 15)' .-the statement s makes little sense and references dont match the statement (14 and 15)

3.line 67- ensure - not assure

4.line 100- better desciption of health facilties and number is needed eg- state total numbr of all the goernment health facilities and priate seperatly, then state the numer used in the study

5.line-121-'included in' not included to th study as written

6. line 118-121- repitatation of line 100- theat section is just for sample size determination

7. line 131- sampling interval (K = N/n) was 2- ...2 can only be for 1 halth facilty , not be same for different study sites with different proportionaly chosen sample sizes and total population

8.line 132- should be ...By balloting not lottery

9.line 197 - should be- fifty eight (20.4%) pregnant women have had abortions ..not ..."Of those pregnant women who have had pregnancy experience, 58 (20.4%) of them have experienced abortion."

10.line 198--200- thirty six(12.9%)Women had experienced pregnancy related complications....,- not ...Women who have faced various pregnancy related complications (pregnancy induced hypertension, anemia, eclampsia, pre-eclampsia) that required hospitalization or additional treatment were 36 (12.9%).

11.line 211- 212 - the women are not reporting , its the knowledge the study tested - so saying One hundred eighty (48%) pregnant women reported that four and above ANC visits are needed for pregnant women during normal pregnancy is not quite right--- rather say 48% had knowledge that (or knew that) pregnant women required four and above ANC visits

12.line 219- 223- how can the women know that they had high Diastolic BP and UTI before ANC???? and started ANC for this cause? Do they have home visits from health workers that test and disgnose them at home before presenting?

13.line 260-261- shoul be- 'The percentage was also higher compared to the finding of xxx% from a systematic review and meta-analysis of studies conducted in Ethiopia'.. not .....The percentage was also higher compared to a systematic review and meta-analysis study conducted in Ethiopia

14.line 277-281- This might be because the more pregnant women know the time of first ANC visit, the more their timely initiation of ANC service will increase ...... this is not an explanation , the authors are stating/repeating the results in a different manner

Reviewer #3: (No Response)

7. PLOS authors have the option to publish the peer review history of their article (what does this mean?). If published, this will include your full peer review and any attached files.

Reviewer #2: No

Reviewer #3: No

---

## [Author Response · Author response to Decision Letter 2]

20 Jul 2022

Response to Reviewers and Editor 

Firstly, we would like to thank the academic editor and reviewers for your additional valuable and constructive comments and questions that are helpful for improving the quality of the manuscript. The responses for each point are done as follows.

To Academic Editor 

• The points raised by reviewer #2 such as clarifications in the reporting, methodology justification, and language/copyediting and the specific issues raised are addressed. 

• A rebuttal letter, Revised Manuscript with Track Changes and Manuscript without track changes are prepared. 

Response to Reviewers

Reviewer #2: 

A. While the authors have attempted to address some of the comments , they failed to work on the general manuscript to improve its intelligibility in standard English as preiuosly advised - this is the major weaknes of this manuscript. They may do well to employ the services of an editor to help with the written language.

• Response: From cover page to last page revision is made to correct writing and grammatical errors as indicated in the tracked changed manuscript. 

B. Explanations for findings are not satisfactory mostly differences were just generic, not much specific explanations of differences between thier findings and the cited works - eg in line 261-266 " The possible reasons for this difference might be due to the difference in accessing information regarding timely initiation of antenatal care visit, advocating the importance of timely initiation of antenatal care, and advising pregnant women on the early booking of ANC, educational status of the pregnant women (14-15, 26). In addition, methodological variations among studies and the difference socio-demographic characteristics of the study participants might be the possible reasons-

• Response: Big revision is made on the explanations (Please see lines 267-280 of the track changed manuscript)

C... specific isssues that should have been worked on and still noted are stated below

1. line 27- should be- late in pregnancy , not in their late pregnancy; check and reword line47; line 50-60

• Response: Changes are made as per the request. (Please see lines 27, 48 and 59 of the track changed manuscript)

2.line 87- the statement -"That is lower than the WHO standard (14, 15)' .-the statement s makes little sense and references dont match the statement (14 and 15)

• Response: Thank you. The statement doesn’t go with the sentences before and after it. So, it is deleted from the manuscript. References 14 and 15 match with the sentence after it and they are included there. (Please see line 87 of the track changed manuscript)

3.line 67- ensure - not assure

• Response: Corrected. (Please see line 67 of the track changed manuscript)

4.line 100- better desciption of health facilties and number is needed eg- state total numbr of all the goernment health facilities and priate seperatly, then state the numer used in the study

• Response: The description is corrected. (Please see lines 101-106 of the track changed manuscript) 

5.line-121-'included in' not included to th study as written

• Response: Corrected. (Please see line 124 of the track changed manuscript)

6. line 118-121- repitatation of line 100- theat section is just for sample size determination

• Response: The repeated part is deleted. (Please see lines 121-124 of the track changed manuscript)

7. line 131- sampling interval (K = N/n) was 2- ...2 can only be for 1 halth facilty , not be same for different study sites with different proportionaly chosen sample sizes and total population

• Response: The sampling interval calculated for each health facility was the same. Since it was proportionally allocated, the sampling interval was the same. It was approximately 2 for each health facility. Modification is done to indicate that the sampling interval was the same for each health facility (Please see line 134 of the track changed manuscript)

8.line 132- should be ...By balloting not lottery

• Response: Corrected (Please see line 135 of the track changed manuscript)

9.line 197 - should be- fifty eight (20.4%) pregnant women have had abortions ..not ..."Of those pregnant women who have had pregnancy experience, 58 (20.4%) of them have experienced abortion."

• Response: Thank you. Corrected as suggested (Please see lines 200-201 of the track changed manuscript)

10.line 198--200- thirty six(12.9%)Women had experienced pregnancy related complications....,- not ...Women who have faced various pregnancy related complications (pregnancy induced hypertension, anemia, eclampsia, pre-eclampsia) that required hospitalization or additional treatment were 36 (12.9%).

• Response: Thank you. Corrected as suggested (Please see lines 203-204 of the track changed manuscript)

11.line 211- 212 - the women are not reporting , its the knowledge the study tested - so saying One hundred eighty (48%) pregnant women reported that four and above ANC visits are needed for pregnant women during normal pregnancy is not quite right--- rather say 48% had knowledge that (or knew that) pregnant women required four and above ANC visits

• Response: Thank you. Corrected as suggested (Please see lines 217-218 of the track changed manuscript)

12.line 219- 223- how can the women know that they had high Diastolic BP and UTI before ANC???? and started ANC for this cause? Do they have home visits from health workers that test and disgnose them at home before presenting?

• Response: This finding was obtained from self-report during the interview. Changes are made to indicate this. (Please see lines 226-229 of the track changed manuscript)

13.line 260-261- shoul be- 'The percentage was also higher compared to the finding of xxx% from a systematic review and meta-analysis of studies conducted in Ethiopia'.. not .....The percentage was also higher compared to a systematic review and meta-analysis study conducted in Ethiopia

• Response: Changes are made (Please see lines 267-270 of the track changed manuscript)

14. line 277-281- This might be because the more pregnant women know the time of first ANC visit, the more their timely initiation of ANC service will increase ...... this is not an explanation , the authors are stating/repeating the results in a different manner

• Response: Changes are made (Please see lines 293-297 of the track changed manuscript)

---

## [Decision Letter · Decision Letter 3]

4 Aug 2022

Timely Initiation of Antenatal Care and Associated Factors among Pregnant Women Attending Antenatal Care in Southwest Ethiopia

PONE-D-21-21466R3

Dear Dr. Bazie,

We’re pleased to inform you that your manuscript has been judged scientifically suitable for publication and will be formally accepted for publication once it meets all outstanding technical requirements.

Kind regards,

Hugh Cowley

Staff Editor

PLOS ONE

Additional Editor Comments (optional):

Reviewers' comments:

Reviewer's Responses to Questions

**Comments to the Author**

1. If the authors have adequately addressed your comments raised in a previous round of review and you feel that this manuscript is now acceptable for publication, you may indicate that here to bypass the “Comments to the Author” section, enter your conflict of interest statement in the “Confidential to Editor” section, and submit your "Accept" recommendation.

Reviewer #2: All comments have been addressed

2. Is the manuscript technically sound, and do the data support the conclusions?

Reviewer #2: Yes

3. Has the statistical analysis been performed appropriately and rigorously? 

Reviewer #2: Yes

4. Have the authors made all data underlying the findings in their manuscript fully available?

Reviewer #2: Yes

5. Is the manuscript presented in an intelligible fashion and written in standard English?

Reviewer #2: Yes

6. Review Comments to the Author

Reviewer #2: The autores have addreeesed the issues i was concerned over, there is no significant problems with legilibilty of the manuscript

7. PLOS authors have the option to publish the peer review history of their article (what does this mean?). If published, this will include your full peer review and any attached files.

Reviewer #2: No

---

## [Editor Report · Acceptance letter]

8 Aug 2022

PONE-D-21-21466R3 

Timely Initiation of Antenatal Care and Associated Factors among Pregnant Women Attending Antenatal Care in Southwest Ethiopia 

Dear Dr. Bazie:

I'm pleased to inform you that your manuscript has been deemed suitable for publication in PLOS ONE. Congratulations! Your manuscript is now with our production department. 

Kind regards, 

on behalf of

Mr Hugh Cowley 

Staff Editor

PLOS ONE